# Intrinsically Guided Exploration in Meta Reinforcement Learning

## Abstract

Deep reinforcement learning algorithms generally require large amounts of data to solve a single task. Meta reinforcement learning (meta-RL) agents learn to adapt to novel unseen tasks with high sample efficiency by extracting useful prior knowledge from previous tasks. Despite recent progress, efficient exploration in meta-training and adaptation remains a key challenge in sparse-reward meta-RL tasks. We propose a novel off-policy meta-RL algorithm to address this problem, which disentangles exploration and exploitation policies and learns intrinsically motivated exploration behaviors. We design novel intrinsic rewards derived from information gain to reduce task uncertainty and encourage the explorer to collect informative trajectories about the current task. Experimental evaluation shows that our algorithm achieves state-of-the-art performance on various sparse-reward MuJoCo locomotion tasks and more complex Meta-World tasks.

## 1 Introduction

Human intelligence is able to transfer knowledge across tasks and acquire new skills within limited experiences. However, most reinforcement learning (RL) agents still require large amounts of data to achieve human-level performance (Silver et al., 2017; Hessel et al., 2018; Vinyals et al., 2019). Meta reinforcement learning (meta-RL) makes a step toward such efficient learning by extracting prior knowledge from a set of tasks to quickly adapt to new tasks. However, efficient exploration of meta-RL needs to consider both training and adaptation phases simultaneously (Ishii et al., 2002), which becomes a key challenge for meta-RL.

One branch of meta-RL algorithms (Finn et al., 2017; Stadie et al., 2018; Rothfuss et al., 2019; Gurumurthy et al., 2019) utilizes policies injected with time-irrelevant random noise for meta-training exploration, while another branch of methods (Duan et al., 2016; Mishra et al., 2017; Gupta et al., 2018; Zintgraf et al., 2019; Rakelly et al., 2019) introduces memories or latent variables that enable temporally-extended exploration behaviors. EPI (Zhou et al., 2018) introduces intrinsic rewards based on the improvement of dynamics prediction. However, these exploration mechanisms are still inefficient in either meta-training or adaptation and underperform in complex sparse-reward tasks.

To address this challenge of meta-RL, we introduce information-theoretic intrinsic motivations for learning to collect informative trajectories and enable efficient exploration in both meta-training and adaptation. Inspired by the common task-inference component in context-based meta-RL algorithms (such as PEARL (Rakelly et al., 2019) and VariBAD (Zintgraf et al., 2019)), we leverage an insight that exploration behaviors should collect trajectories that contain rich information gain about the current task, and design an exploration objective to maximize the information gain for inferring taskss. Based this objective, we derive an intrinsic reward for learning an effective exploration policy. To reduce variance from estimating information-gain intrinsic rewards in complex domains, we simplify and derive an intrinsic reward based on prediction errors to achieve superior stability and scalability.

We develop a novel off-policy meta-RL algorithm, Meta-RL with effiCient Uncertainty Reduction Exploration (MetaCURE), that incorporates our intrinsic rewards and separates exploration and exploitation policies. MetaCURE learns to perform sequential exploration behaviors to reduce task uncertainty across adaptation episodes and maximizes the expected extrinsic return in the last episode of the adaptation phase. During meta-training, MetaCURE collects training data from both exploration and exploitation policies. As for adaptation, the exploration policy collects informative

trajectories, and then the exploitation policy utilizes gained experiences to maximize final performance. We evaluate our algorithm on various sparse-reward MuJoCo locomotion tasks as well as sparse-reward Meta-World tasks. Empirical results show that it outperforms baseline algorithms by a large margin. We also visualize how our algorithm explores in novel tasks and discuss the pros and cons of the two proposed intrinsic rewards.

## 2 BACKGROUND

In meta-RL, we consider a distribution of tasks $p(\mathcal{T})$, with each task $\mathcal{T}$ modelled as a Markov Decision Process (MDP), which consists of a state space, an action space, a transition function and a reward function. In common meta-RL settings (Duan et al., 2016; Finn et al., 2017; Zintgraf et al., 2019; Rakelly et al., 2019), tasks differ in the transition and/or reward function, and we can describe a task $\mathcal{T}$ with a tuple $\langle p_0^{\mathcal{T}}(s_0), p^{\mathcal{T}}(s'|s,a), r^{\mathcal{T}}(s,a) \rangle$, with $p_0^{\mathcal{T}}(s_0)$ the initial state distribution, $p^{\mathcal{T}}(s'|s,a)$ the transition probability and $r^{\mathcal{T}}(s,a)$ the reward function.

We denote context $c_n^{\mathcal{T}} = (a_n, r_n, s_{n+1})$ as the experience tuple collected at the $n$-th step of adaptation in task $\mathcal{T}$, and we use $c_{-1:T-1}^{\mathcal{T}} = \langle c_{-1}^{\mathcal{T}}, c_0^{\mathcal{T}}, ..., c_{T-1}^{\mathcal{T}} \rangle^1$ to denote all trajectories collected in the T timesteps

A common objective for meta-RL is to optimize final performance after few-shot adaptation (Finn et al., 2017; Gupta et al., 2018; Stadie et al., 2018; Rothfuss et al., 2019). During adaptation, an agent first utilizes some exploration policy $\pi^e$ to explore for a few episodes, and then updates an exploitation policy $\pi$ to maximize expected return. Such a meta-RL objective can be formulated as:

$$\max_{\pi, \pi^e} \mathbb{E}_{\mathcal{T}}[R(\mathcal{T}, \pi(c_{\pi^e}^{\mathcal{T}}))] \tag{1}$$

where $c_{\pi^e}^{\mathcal{T}}$ is a set of experiences collected by $\pi^e$, and $R$ is the last episode's expected return with $\pi$. The exploitation policy $\pi$ is adapted with $c_{\pi^e}^{\mathcal{T}}$ for optimizing final performance.

## 3 METACURE

To support efficient exploration in both meta-training and adaptation, we propose MetaCURE, a novel off-policy meta-RL algorithm, that learns separate exploration and exploitation policies. The exploration policy aims to collect trajectories that maximize the agent's information gain to reduce uncertainty of task inference. The exploitation policy maximizes the expected extrinsic return in the last episode of adaptation. In this section, we first present the MetaCURE framework and then discuss its intrinsic reward design for learning an efficient exploration policy for both meta-training and adaptation.

### 3.1 THE METACURE FRAMEWORK

As shown in Figure 1, MetaCURE is composed of three main components: (i) a task encoder $q_\phi(z|c)$ that extracts information from context $c$ and estimates the posterior of the task belief $z$, (ii) an *Explorer* to learn an behavior or exploration policy, and (iii) an *Exploiter* to learn the target or exploitation policy. We utilize variational inference methods (Kingma & Welling, 2013; Alemi et al., 2016; Rakelly et al., 2019) to train the task encoder $q_\phi$. In order to learn effective task encodings, its decoder is designed to recover the action value function of the exploiter, which captures rich and temporally-extended information about the current task.

The algorithm utilizes two separate replay buffers, $B$ and $B_{enc}$. Buffer $B$ is used to train both the *Explorer* and the *Exploiter*, while buffer $B_{enc}$ is used to train the task encoder. During meta-training, MetaCURE iteratively infers the posterior task belief with contexts and performs both exploration and exploitation policies to collect data. $B$ stores experiences from both policies, while $B_{enc}$ only stores exploration trajectories. During the adaptation phase, only the explorer is used to collect informative trajectories for inferring the posterior task belief, and the exploiter utilizes this posterior belief to maximize final performance.

---

[1]for $n = -1$, we define $c_{-1}^{\mathcal{T}} = (\vec{0}, \vec{0}, s_0)$. In following derivations, we may drop $\mathcal{T}$ for brevity.

To learn efficient exploration in sparse-reward tasks, the reward signal for the explorer is defined as follows:

$$r_{\text{explorer}}(c_{-1:i-1}, a_i) = \gamma^n r(s_i, a_i) + \lambda r'_{intr}(c_{-1:i-1}, a_i) \tag{2}$$

where $r$ is the extrinsic reward, $r'_{intr}$ is an intrinsic reward, $\lambda > 0$ is a hyper-parameter and $\gamma \in [0, 1]$ is a discount factor, and $n$ is the number of training iterations. The intrinsic reward is conditioned on past trajectories $c$, which direct the explorer to collect relevant information for task inference. Empirically, we utilize the task belief $q_\phi(z|c)$ instead of context $c$ as explorer's input, because it extracts useful task-specific information. This modification reuses learned knowledge from raw experiences and accelerates learning. We will discuss in detail the design of intrinsic reward $r'_{intr}$ in the next section. As training proceeds, the extrinsic reward gradually diminishes, and the explorer converges to behaviors that collects useful experiences for adaptation.

The exploiter $\pi$ is expected to utilize collected experiences and optimize exploitation behavior. We design it to take state $s$ and the latent code $z$ as input, and maximize the expected extrinsic return. Finally, we utilize SAC (Haarnoja et al., 2018), an off-policy RL algorithm to train the policies. Algorithm pseudo-codes are available in Appendix A.3.

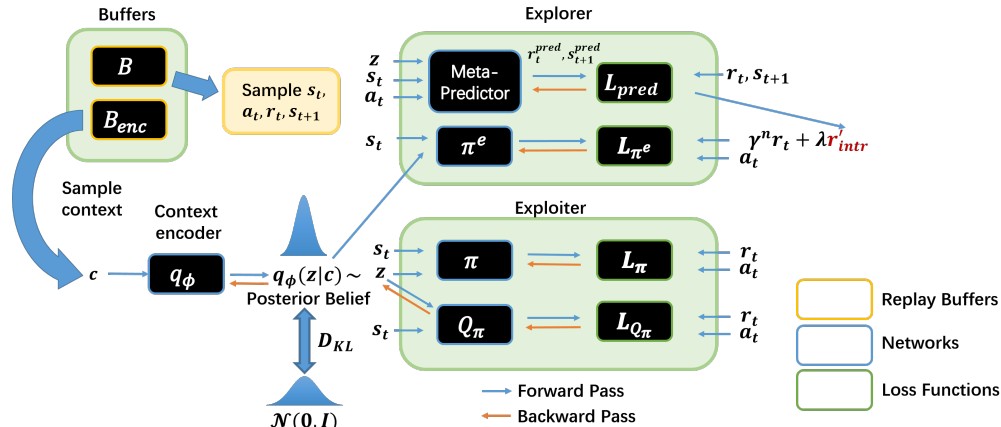

Figure 1: MetaCURE's meta-training pipeline (with prediction error intrinsic). Sharing buffers for two policies achieves high sample efficiency as both policies can be off-policy trained with history experiences. Batches from $B$ and $B_{enc}$ are sampled from the same task. $\pi^e$'s Q function is omitted for brevity.

## 3.2 INTRINSIC REWARDS FOR METACURE

As shown in Figure 1, to support efficient exploration in both meta-training and adaptation, we maximize agent's information gain during exploration, and introduce an information-theoretic intrinsic reward in the following form:

$$r'_{IG}(c_{-1:i-1}, a_i) = \mathbb{E}_{(r_i, s_{i+1})|(c_{-1:i-1}, a_i)}[D_{KL}[p(z|c_{-1:i-1})||p(z|c_{-1:i})]] \tag{3}$$

The left-hand term $p(z|c_{-1:i-1})$ in the KL distance is the agent's task belief before observing $c_i$, while the right-hand term is the posterior task belief. This term reflects how much the agent's belief has changed after collecting an experience. We also observe similar intrinsic rewards proposed in VIME (Houthooft et al., 2016), in which they explain the rewards as compression improvement.

Further derivation (see Appendix A.1 for details) shows that this intrinsic reward can be estimated by the difference of two prediction errors:

$$
\begin{aligned}
&r'_{IG}(c_{-1:i-1}, a_i) \\
&= \mathbb{E}_{(r_i, s_{i+1})|(c_{-1:i-1}, a_i)}[D_{KL}[p(z|c_{-1:i-1})||p(z|c_{-1:i})]] \\
&= \mathbb{E}_{(z, r_i, s_{i+1})|(c_{-1:i-1}, a_i)}[\log p(r_i, s_{i+1}|c_{-1:i-1}, a_i) - \log p(r_i, s_{i+1}|c_{-1:i-1}, a_i, z)]
\end{aligned} \tag{4}
$$

Although sharing some similarities with EPI (Zhou et al., 2018), our intrinsic reward possesses several key differences: first, it maximizes both information gains in reward and dynamics prediction, while EPI ignores reward signals that are often critical for task inference in meta-RL; secondly, our intrinsic reward supports end-to-end learning and collect online data for effective task inference, while EPI requires a fixed dataset to compute its intrinsic reward and lacks an effective mechanism for adaptive exploration.

Empirically, we find that this intrinsic reward suffers form high variance, sometimes causing instability in training. Inspired by curiosity-driven exploration in traditional RL (Still & Precup, 2012; Pathak et al., 2017; Burda et al., 2018), we remove the first term in $r'_{IG}$ and use prediction errors as the intrinsic reward to reduce the estimation variance:

$$r'_{PE}(c_{-1:i-1}, a_i) = \mathbb{E}_{(z, r_i, s_{i+1})|(c_{-1:i-1}, a_i)}[-\log p(r_i, s_{i+1}|c_{-1:i-1}, a_i, z)] \tag{5}$$

Unlike the prediction-based curiosity in traditional RL, this intrinsic reward is estimated on multiple tasks, and does not diminish during meta-training.

## 4  EXPERIMENTS

In this section, we aim at answering the following questions: 1. Can MetaCURE achieve good final adaptation performance in sparse-reward tasks that require efficient exploration in both meta-training and adaptation? 2. Do the explorer and exploiter obtain desirable behaviors? 3. What's the pros and cons of the two proposed intrinsic rewards? 4. Is MetaCURE's components vital for efficient exploration? Besides, we are also interested in its sample efficiency in meta-training and adaptation.

### 4.1  ADAPTATION PERFORMANCE ON CONTINUOUS CONTROL

**Environment Setup** We evaluate our algorithm on various continuous control task sets with sparse rewards, in which exploration is vital for good performance. These tasks vary either in the reward function (goal location in Point-Robot-Sparse and Reacher-Goal-sparse, target velocity in Cheetah-Vel-Sparse and Walker-Vel-Sparse) or the transition function (Walker-Params-Sparse). These tasks (except for Point-Robot-Sparse) are simulated via MuJoCo (Todorov et al., 2012) and are benchmarks commonly used by current meta-learning algorithms (Mishra et al., 2017; Finn et al., 2017; Rothfuss et al., 2019; Rakelly et al., 2019). Unlike previous evaluation settings, we limit the length of adaptation phase to address the importance of efficient adaptation exploration. Also, we do not provide dense rewards while meta-training, which is different from the setting of PEARL. Typically, each adaptation phase consists of 2∼4 episodes, each episode 32∼64 steps, varying with the specific task set. Detailed parameters and reward function settings are available in Appendix A.4.

**Algorithm setup** We compare our algorithm against several representative meta-RL algorithms, including RL$^2$ (Duan et al., 2016), MAML (Finn et al., 2017), E-RL$^2$ (Stadie et al., 2018), ProMP (Rothfuss et al., 2019), MAME (Gurumurthy et al., 2019), PEARL (Rakelly et al., 2019) and VariBAD (Zintgraf et al., 2019). We also compare with EPI (Zhou et al., 2018), and modify it to make it consider reward predictions[2]. We use open-source codes provided by the original papers, and the performance is averaged over three random seeds. We implement two variants of MetaCURE with PyTorch (Paszke et al., 2019): MetaCURE-IG utilizes the information-gain intrinsic in Equation 4, and MetaCURE-PE utilizes the prediction error intrinsic in Equation 5. For both variants, the exploitation policy is only performed in the last adaptation episode, and the exploration policy is used for other adaptation episodes.

**Results and analyses** As shown in Figure 2 and Appendix A.5, we plot the algorithms' meta-testing performance as a function of number of experiences collected during meta-training. Performance is evaluated by the last adaptation episode's return averaged over all meta-testing tasks. MetaCURE is the only algorithm that manages to acquire good performance on all five task sets, and MetaCURE with the prediction error intrinsic performs slightly better. PEARL fails to effectively explore within

---

[2]EPI is not trained end-to-end,and we plot its final performance in dash line.

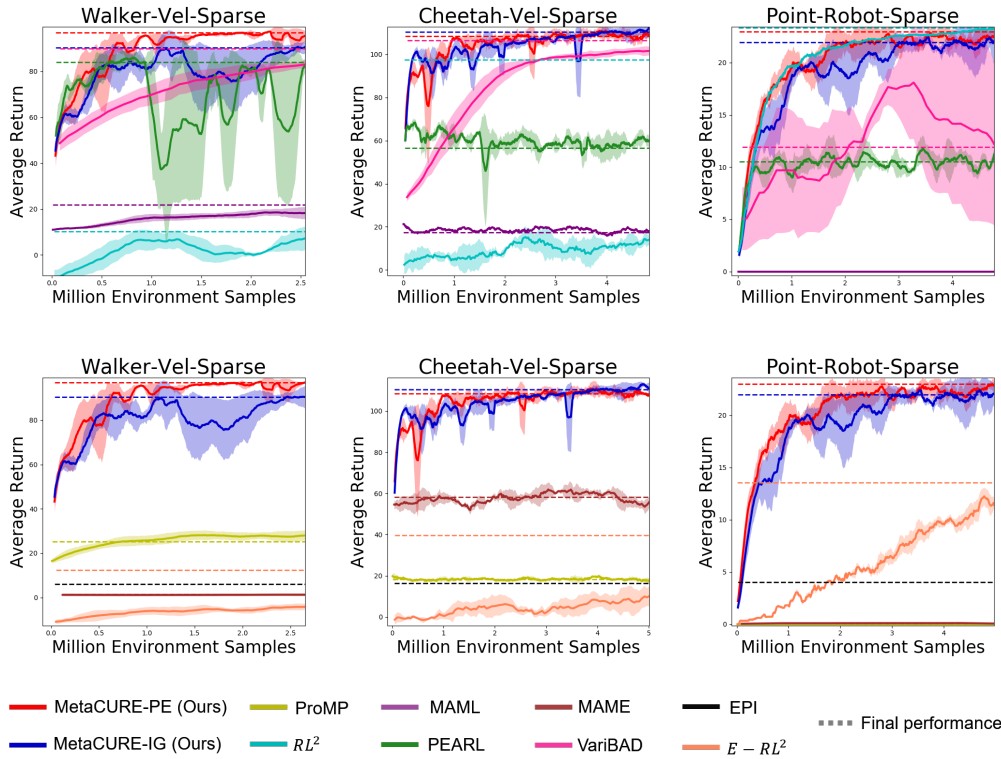

Figure 2: Evaluation of MetaCURE and several meta-RL baselines on various sparse-reward continuous control task sets. MetaCURE achieves superior final performance as well as high sample efficiency, while none of the baseline algorithms manage to solve all five tasks. MetaCURE-PE slightly outperforms MetaCURE-IG.

short adaptation, as posterior sampling (Osband & Van Roy, 2017) is only optimal in asymptotic performance. $RL^2$ achieves good performance in tasks with simple dynamics like Point-Robot-Sparse and Cheetah-Vel-Sparse, but fails in more complex tasks. MAML acquires good performance in Walker-Params-Sparse but fails to learn tasks with different reward functions. The rest of the baselines also perform poorly with sparse rewards. As for sample efficiency, MetaCURE and PEARL enjoy higher sample efficiency as they are off-policy trained, needing far fewer samples to converge.

## 4.2 ADAPTATION PERFORMANCE ON META-WORLD

Meta-World (Yu et al., 2019) is a recently proposed challenging evaluation benchmark for meta-RL, including a variety of robot arm control tasks. We test MetaCURE-PE as well as baselines on two Meta-World task sets: Push and Reach. Although these task sets are not very distinct, they are still very difficult to learn due to their complexity in dynamics and long time-horizon. To address the importance of exploration, we make the rewards sparse, which makes the tasks even harder. Following the setting in Yu et al. (2019), we evaluate algorithms by their final success rates, and results are shown in Table 1. MetaCURE achieves significantly higher success rates than baselines by achieving efficient exploration.

## 4.3 ADAPTATION VISUALIZATION

To prove that MetaCURE indeed learns efficient exploration and exploitation strategies, we visualize MetaCURE-PE's adaptation phase in Point-Robot-Sparse and Walker-Vel-Sparse, as shown in Figure 3 and Figure 4. We compare against PEARL, which explores via posterior sampling. While MetaCURE effectively explores before carrying out good exploitation behaviors, PEARL keeps its

Table 1: Final success rates on the hard Meta-World tasks.

| ENVIRONMENTS | METACURE-PE | PEARL | MAML | PROMP | RL$^2$ | E-RL$^2$ |
|---|---|---|---|---|---|---|
| META-WORLD PUSH | **0.46±0.10** | 0.28±0.11 | 0±0 | 0±0 | 0±0 | 0±0 |
| META-WORLD REACH | **0.25±0.07** | 0.07±0.04 | 0±0 | 0±0 | 0±0 | 0±0 |

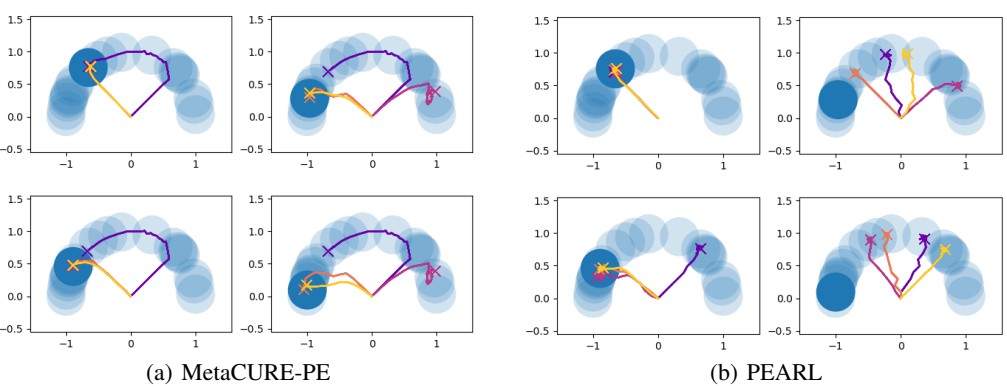

(a) MetaCURE-PE        (b) PEARL

Figure 3: Adaptation visualization of MetaCURE-PE and PEARL on Point-Robot-Sparse. The agent is given four episodes to perform adaptation. Trajectories in dark purple are trajectories of the first adaptation episode, while the light yellow trajectories are trajectories of the last episode. Dark blue circles indicate rewarding regions for current task, while light blue circles indicate rewarding regions for other meta-testing tasks. PEARL explores by rolling out exploitation policies. MetaCURE-PE is intrinsically motivated to gather task-specific knowledge, efficiently exploring possible goals and achieving good exploration and exploitation behaviors.

belief about current task throughout an entire episode, and explores less effectively. We additional visualize adaptation in Cheetah-Vel-Sparse, and results are available in Appendix A.6.

## 4.4 INFORMATION-GAIN VERSUS PREDICTION ERROR

In Figure 2, MetaCURE-PE outperforms MetaCURE-IG, as its intrinsic reward is of lower variance. However, the prediction errors may lead to bad exploration behaviors under certain scenarios. In this subsection we discuss about irrelevant dynamics, a problem that MetaCURE-PE may fail to solve while MetaCURE-IG manages to deal with.

We discuss a 2D navigation task set called Point-Robot-Sparse-Noise. The agent's state is a 3D vector composed of its current position $(x, y)$ and a random noise $z$, which is sampled from a uniform distribution $[0, d]$, with $d$ the L2 distance between the agent's current position and the original point. Goals are uniformly distributed on a semicircle of radius 1 and only sparse reward is provided. We test MetaCURE with two intrinsic rewards on this task set, and performance and visualizations are shown in Figure 5. While MetaCURE-PE is interested in distinct regions where the prediction loss is large and fails to explore possible goals, MetaCURE-IG manages to effectively collect information about task and ignores irrelevant noises in dynamics. We also provide another example in Appendix A.7.

## 4.5 ABLATION STUDY

In this section, we discuss the necessity of MetaCURE's components for its efficient exploration.

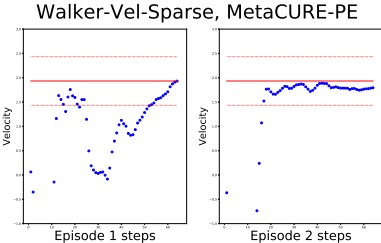
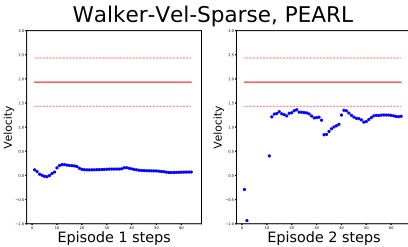

Figure 4: Visualization of MetaCURE-PE and PEARL on the Walker-Vel-Sparse tasks. The solid red line is the target velocity, and the region bounded by red dash lines represents velocities that get informative rewards. While PEARL keeps its belief unchanged during an entire episode, MetaCURE-PE first efficiently explores the goal velocity before performing good exploitation policies.

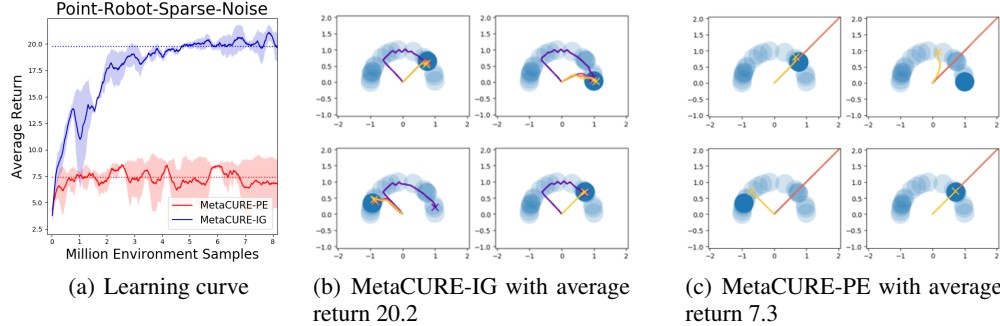

| (a) Learning curve | (b) MetaCURE-IG with average return 20.2 | (c) MetaCURE-PE with average return 7.3 |

Figure 5: (a) Learning curves of MetaCURE-IG and MetaCURE-PE on Point-Robot-Sparse-Noise. (b) Adaptation visualization of MetaCURE-IG. It is motivated to gain task information, and is not disturbed by noises in dynamics. (c) Adaptation visualization of MetaCURE-PE. It fails to obtain good exploration behaviors, as distinct regions obtain nosier dynamics and larger prediction losses.

**Hyper-parameters** We conduct ablation studies on MetaCURE's hyper-parameters, and results are shown in Appendix A.8. Results show that MetaCURE is generally robust to hyper-parameters and does not need careful fine-tuning.

**Intrinsic rewards** The intrinsic rewards are vital for exploration in both meta-training and adaptation. MetaCURE without intrinsic rewards cannot explore effectively during meta-training, as shown in Appendix A.9.1. We also show algorithms' performance on another task set, in which adaptation is short and the agent cannot perform good adaptation exploration by maximizing average return during adaptation. As shown in Appendix A.9.2, while MetaCURE with intrinsic rewards is able to perform good exploration behaviors that benefit final adaptation performance, MetaCURE without intrinsic rewards is attracted by the sub-optimal goal and cannot acquire necessary information for final exploitation.

**Baselines with intrinsic rewards** We additionally test baselines that simply add our intrinsic reward (curiosity intrinsic) to PEARL and E-RL[2], respectively (for E-RL[2], the intrinsic is provided to the exploration episodes only). On the Cheetah-Vel-Sparse task set, while MetaCURE achieves a performance of $104.5 \pm 2.2$, PEARL with intrinsic rewards only achieves $96.6 \pm 4.5$, and E-RL[2] with intrinsic rewards only achieves $92.3 \pm 9.3$. These results show that trivially combining our intrinsic rewards with meta-RL does not lead to good performance. MetaCURE outperforms these two baselines by performing task inference and separating exploration and exploitation.

## 5 RELATED WORKS

**Exploration in meta-RL** Compared to meta supervised learning, in meta-RL the agent is not given a task-specific dataset to adapt to, and it must explore the environment to collect useful information. This exploration part is vital for both meta-training and adaptation (Finn & Levine, 2019).

Adaptation exploration in gradient-based meta-RL is mainly addressed by computing the gradient to the pre-update policy's state distribution (Stadie et al., 2018; Rothfuss et al., 2019). MAME (Guru-murthy et al., 2019) learns a separate explorer, and adapts via self-supervised losses. These methods utilize policy injected with random noise for meta-training exploration, which is ineffective and empirically fail with sparse rewards. MAESN (Gupta et al., 2018) introduces temporally-extended exploration during adaptation with latent variables. Most context-based methods automatically balances between exploration and exploitation during adaptation by maximizing average adaptation performance (Duan et al., 2016; Zintgraf et al., 2019), and E-RL$^2$ (Stadie et al., 2018) directly optimizes for final performance. These methods also fail to meta-train policies with sparse rewards. PEARL (Rakelly et al., 2019), a recently proposed approach, utilizes posterior sampling (Osband & Van Roy, 2017) to explore, and is less efficient with short adaptation, as posterior sampling is only optimal is asymptotic performance. EPI (Zhou et al., 2018) learns exploration policies guided with intrinsic rewards based on prediction improvement in dynamics. However, it ignores useful reward signals, and lacks efficient exploration mechanisms to collect the dataset for intrinsic reward estimation, especially in sparse-reward settings.

**Exploration with information-theoretic intrinsic rewards** Measuring information gain is an important way of designing intrinsic rewards for exploration. VIME (Houthooft et al., 2016) measures the mutual information between trajectories and the transition function, while EMI (Kim et al., 2019) measures the mutual information between state and action with the next state in consideration, as well as the mutual information between state and next state with the action in consideration, both in the latent space. Sun et al. (2011) discusses exploration with information gain, but is restricted to planning problems and requires an oracle estimating the posterior.

**Prediction loss as intrinsic reward** Prediction loss can serve as a kind of curiosity-driven intrinsic reward to encourage exploration. Intuitively, high prediction loss implies that the situation has not been fully explored, and visiting these situations helps the agent to explore. Oh et al. (2015) directly predicts the image observation, and Stadie et al. (2015) utilize prediction loss in the latent space, extracting useful features from observations. To avoid trivial solutions in learning the latent space, Pathak et al. (2017) introduces an inverse model to guide the learning of latent space, only predicting things that the agent can control. Burda et al. (2018) utilizes random neural networks as projections onto the latent space and achieved good performance on Atari games.

## 6 CONCLUSION

In this paper, we propose a novel off-policy meta-RL algorithm to facilitate meta-training and adaptation exploration, which becomes critical for sparse reward signals. By introducing information-theoretic intrinsic motivations and separating exploration and exploitation policies, our method is able to acquire sophisticated exploration behaviors. Our algorithms of two intrinsic rewards both achieve the state-of-the-art performance on various sparse-reward MuJoCo locomotion task sets and more difficult Meta-World tasks.

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

## A APPENDIX

### A.1 PROOF OF EQUATION 4

We prove the following equation:

$$
\begin{aligned}
&\mathbb{E}_{(r_i, s_{i+1})|(c_{-1:i-1}, a_i)}[D_{KL}[p(z|c_{-1:i-1})||p(z|c_{-1:i})]] = \\
&\mathbb{E}_{(z, r_i, s_{i+1})|(c_{-1:i-1}, a_i)}[\log p(r_i, s_{i+1}|c_{-1:i-1}, a_i) - \log p(r_i, s_{i+1}|c_{-1:i-1}, a_i, z)]
\end{aligned}
\tag{6}
$$

*Proof.* We first state an important fact before providing our proof. *Knowing an action without knowing subsequent observations and rewards does not bring information about the task, or say:*

$$p(z|c_{-1:i-1}) = p(z|c_{-1:i-1}, a_i) \tag{7}$$

This fact is also used in Sun et al. (2011) and Houthooft et al. (2016). Then

$$
\begin{aligned}
&\mathbb{E}_{(r_i, s_{i+1})|(c_{-1:i-1}, a_i)}[D_{KL}[p(z|c_{-1:i-1})||p(z|c_{-1:i})]]\\
&= \iiint p(z|c_{-1:i-1})p(r_i, s_{i+1}|c_{-1:i-1}, a_i) \log \frac{p(z|c_{-1:i-1})p(c_i|c_{-1:i-1}, a_i)}{p(z|c_{-1:i})p(c_i|c_{-1:i-1}, a_i)} dr_i ds_{i+1} dz\\
&= \mathbb{E}_{z|c_{-1:i-1}}[\iint p(r_i, s_{i+1}|c_{-1:i-1}, a_i) \log \frac{p(z|c_{-1:i-1})p(c_i|c_{-1:i-1}, a_i)}{p(z, c_i|c_{-1:i-1}, a_i)} dr_i ds_{i+1}]\\
&= \mathbb{E}_{z|c_{-1:i-1}}[\iint p(r_i, s_{i+1}|c_{-1:i-1}, a_i) \log \frac{p(c_i|c_{-1:i-1}, a_i)}{p(c_i|c_{-1:i-1}, a_i, z)} dr_i ds_{i+1}]\\
&= \mathbb{E}_{z|c_{-1:i-1}}[D_{KL}[p(r_i, s_{i+1}|c_{-1:i-1}, a_i)||p(r_i, s_{i+1}|c_{-1:i-1}, a_i, z)]]\\
&= \mathbb{E}_{(z, r_i, s_{i+1})|(c_{-1:i-1}, a_i)}[\log p(r_i, s_{i+1}|c_{-1:i-1}, a_i) - \log p(r_i, s_{i+1}|c_{-1:i-1}, a_i, z)]
\end{aligned}
\tag{8}
$$

One may concern that in Section 3.1, the explorer takes $q_\phi(z|c)$ instead of $c$ as input, and may violate the fact proposed above, as the action may obtain extra information about $z$. However, actually $q_\phi(z|c)$ is a Gaussian distribution parameterized by $\mu_z$ and $\sigma_z$, both of which are deterministic functions of $c$. So knowing $q_\phi(z|c)$ does not introduce extra information about $z$, and does not violate the proposed fact.

## A.2 ESTIMATING INFORMATION-THEORETIC INTRINSIC REWARDS

The intrinsic rewards proposed in Equation 4 and Equation 5 requires estimating density functions $\log p(r_i, s_{i+1}|c_{-1:i-1}, a_i, z)$ and $\log p(r_i, s_{i+1}|c_{-1:i-1}, a_i)$, which is difficult in continuous state and action spaces. In implementation, we estimate them with prediction errors, which is commonly used in density estimation (Chung et al., 2015; Babaeizadeh et al., 2017; Ratzlaff et al., 2019):

$$
\begin{aligned}
&r'_{IG}(c_{-1:i-1}, a_i)\\
&\approx \mathbb{E}_{(z, r_i, s_{i+1})|(c_{-1:i-1}, a_i)}[|(r_i, s_{i+1}) - (r_{pred}(c_{-1:i-1}, a_i, z), s_{pred}(c_{-1:i-1}, a_i, z))|_2\\
&- |(r_i, s_{i+1}) - (r_{pred}(c_{-1:i-1}, a_i), s_{pred}(c_{-1:i-1}, a_i))|_2]
\end{aligned}
\tag{9}
$$

$$
\begin{aligned}
&r'_{PE}(c_{-1:i-1}, a_i)\\
&\approx \mathbb{E}_{(z, r_i, s_{i+1})|(c_{-1:i-1}, a_i)}[|(r_i, s_{i+1}) - (r_{pred}(c_{-1:i-1}, a_i, z), s_{pred}(c_{-1:i-1}, a_i, z))|_2]
\end{aligned}
\tag{10}
$$

## A.3 ALGORITHM PSEUDO-CODES

In this subsection we provide pseudo-codes for MetaCURE's meta-training and adaptation, as shown in Algorithm 1 and Algorithm 2.

## A.4 ENVIRONMENT AND HYPER-PARAMETER SETTINGS

In this subsection, we provide detailed settings of reward functions and hyper-parameters, which are shown in Table 2 and 3. All tasks obtain sparse reward functions, providing zero rewards if the agent is outside the range of goals. The agent is also penalized with control costs.

As for the Meta-World tasks, adaptation phase consists of 4 episodes, with 150 steps in each episode. The agent gets non-zero rewards only if it "successes" in the task, which is given as a binary signal by the environment.

## A.5 ADAPTATION PERFORMANCE

In this subsection we plot learning curves on Reacher-Goal-Sparse and Walker-Rand-Params, as shown in Figure 6.

---

**Algorithm 1** MetaCURE: Meta-training Phase

---

**Input:** A set of meta-training tasks $\{\mathcal{T}^i\}_{i=1,2,...,T}$ drawn from $p(\mathcal{T})$
Initialize replay buffers $\mathcal{B}^i, \mathcal{B}^i_{enc}$ for each task $\mathcal{T}^i$
**while** not done **do**
   **for** each task $\mathcal{T}^i$ **do**
      Initialize experience context $c_i = \{\}$
      **for** episodes=1,...,$K_1$ **do**
         **for** steps=1,...,$H$ **do**
            Take action according to $\pi^e(a|s, q_\phi(z|c_i))$
            Add collected experience $(s, a, r, s')$ to $c_i$, $\mathcal{B}^i$ and $\mathcal{B}^i_{enc}$
         **end for**
      **end for**
      **for** episodes=1,...,$K_2$ **do**
         Sample $z \sim q_\phi(z|c_i)$
         Collect a trajectory with $\pi(a|s, z)$, add collected experience to $\mathcal{B}^i$
      **end for**
   **end for**
   **for** steps in training steps **do**
      **for** each $\mathcal{T}^i$ **do**
         Sample encoder batch $b^i_{enc}$ and policy training batch $b^i$
         Sample $z \sim q_\phi(z|b^i_{enc})$, compute $r^e$ with Equation 5
         Train encoder $q_\phi$, $\pi$ and $\pi^e$ using SAC, $b^i$ and $r^e$, train the predictor to minimize prediction loss
      **end for**
   **end for**
**end while**

---

**Algorithm 2** MetaCURE: Adaptation Phase

---

**Input:** Meta-test task $\mathcal{T}$ drawn from $p(\mathcal{T})$
Initialize experience context $c = \{\}$
**for** episodes=1,...,$N - 1$ **do**
   **for** steps=1,...,$H$ **do**
      Take action according to $\pi^e(a|s, q_\phi(z|c))$
      Add collected experience $(s, a, r, s')$ to $c$
   **end for**
**end for**
Sample $z \sim q_\phi(z|c)$
**for** steps=1,2,...,$H$ **do**
   Take action according to $\pi(a|s, z)$
**end for**

---

### A.6 VISUALIZATIONS

We show additional visualization results on Cheetah-Vel-Sparse, as shown in Figure 7.

### A.7 ANOTHER EXAMPLE OF IRRELEVANT DYNAMICS

In this subsection we discuss another task set that MetaCURE-PE fails on, called Sparse-Point-Robot-Random. In this task set, there exists another kind of noise that is irrelevant to optimal policies: if the agent get close to $(0, -1)$, the agent will be transmitted to a point randomly sampled on a circle of radius 5. As these points are faraway from possible goals and does not provide any task-specific knowledge, optimal policies will never get close to $(0, -1)$.

Algorithms' performance is shown in Figure 8. As MetaCURE-IG insists on probing task-specific information, MetaCURE-PE keeps going to the green circle to get transmitted, and fails to gain useful knowledge for adaptation.

Table 2: Adaptation length and goal settings for environments used to evaluate our algorithm. Goals are uniformly distributed in *Goal range* and non-zero informative reward is provided only when the distance between the agent's position/speed and the goal is smaller than *Goal radius*.

| ENVIRONMENT | # OF ADAPTATION EPISODES | MAX STEPS PER EPISODE | GOAL TYPE | GOAL RANGE | GOAL RADIUS |
|---|---|---|---|---|---|
| CHEETAH-VEL-SPARSE | 2 | 64 | VELOCITY | [0,3] | 0.5 |
| WALKER-VEL-SPARSE | 2 | 64 | VELOCITY | [0,2] | 0.5 |
| REACHER-GOAL-SPARSE | 2 | 64 | POSITION | SEMICIRCLE WITH RADIUS 0.25 | 0.09 |
| POINT-ROBOT-SPARSE | 4 | 32 | POSITION | SEMICIRCLE WITH RADIUS 1 | 0.3 |
| WALKER-PARAMS-SPARSE | 4 | 64 | VELOCITY | 1.5 | 0.5 |

Table 3: Hyperparameter settings for MetaCURE on these tasks. We use 3e-4 for all learning rates.

| ENVIRONMENT | LATENT SIZE | $\beta$ | $\lambda$ |
|---|---|---|---|
| CHEETAH-VEL-SPARSE | 2 | 0.1 | 0.1 |
| WALKER-VEL-SPARSE | 2 | 1 | 0.2 |
| REACHER-GOAL-SPARSE | 5 | 1 | 3 |
| POINT-ROBOT-SPARSE | 4 | 1 | 3 |
| WALKER-PARAMS-SPARSE | 2 | 1 | 3 |
| META-WORLD PUSH | 4 | 0.1 | 3 |
| META-WORLD REACH | 4 | 0.1 | 3 |

## A.8 HYPER-PARAMETER ABLATION STUDIES

We test MetaCURE-PE with a wide range of hyper-parameters on the Cheetah-Vel-Sparse task set, and results are shown in Table 4. $\beta$ is a hyper-parameter controlling the weight of KL divergence in the encoder's loss, similar to Rakelly et al. (2019).

## A.9 ABLATION STUDY: INTRINSIC REWARDS

### A.9.1 INTRINSIC REWARDS FOR META-TRAINING EXPLORATION

We test MetaCURE without intrinsic rewards on Point-Robot-Sparse, and results are shown in Figure 9. Without intrinsic rewards, MetaCURE cannot perform good meta-training exploration and leads to poor performance.

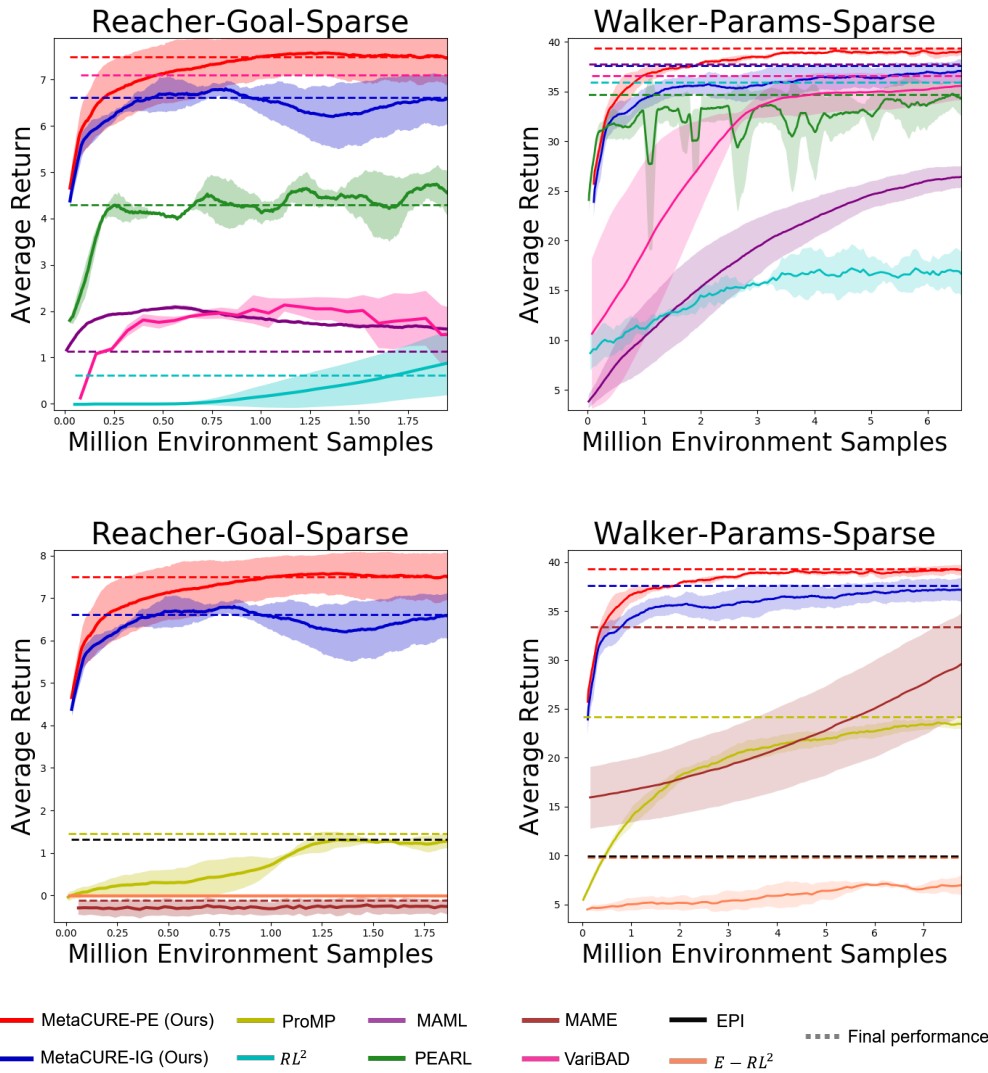

Figure 6: Evaluation of MetaCURE and several meta-RL baselines on Reacher-Goal-Sparse and Walker-Rand-Params.

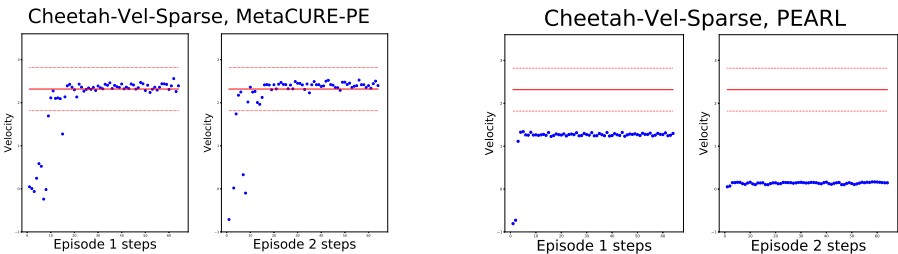

Figure 7: Visualization of MetaCURE and PEARL on the Cheetah-Vel-Sparse tasks.

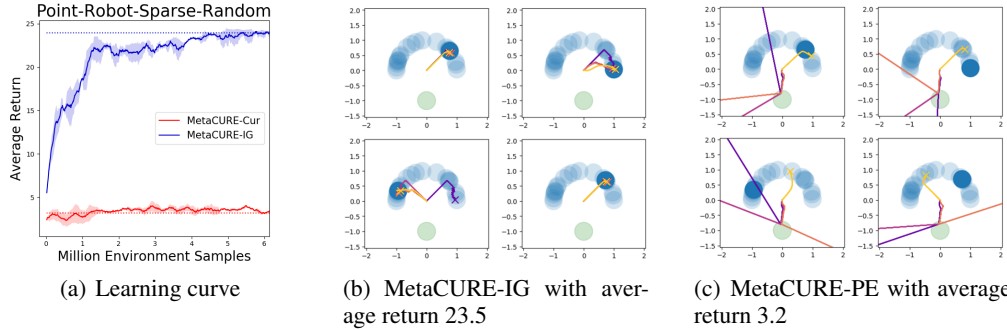

(a) Learning curve      (b) MetaCURE-IG with average return 23.5      (c) MetaCURE-PE with average return 3.2

Figure 8: (a) Learning curves of MetaCURE-IG and MetaCURE-PE on Point-Robot-Sparse-Random. (b) Adaptation visualization of MetaCURE-IG. The agent gets transmitted when it touches the green circle. (c) Adaptation visualization of MetaCURE-PE. It fails to obtain good exploration behaviors, as it is interested in the randomness of being transmitted.

Table 4: MetaCURE's hyper-parameter ablation studies on the Cheetah-Vel-Sparse task set.

| LEARNING RATE | $\beta$ | $\lambda$ | PERFORMANCE |
|---|---|---|---|
| 3E-4 | 1E-1 | 1E-1 | 104.5±2.2 |
| 1E-3 | 1E-1 | 1E-1 | 106.8±6.5 |
| 3E-4 | 1E-2 | 1E-1 | 105.5±3.7 |
| 3E-4 | 1E-1 | 5E-2 | 107.9±6.2 |
| 3E-4 | 1E-2 | 5E-2 | 105.8±2.7 |

### A.9.2 INTRINSIC REWARDS FOR ADAPTATION EXPLORATION

To show how intrinsic rewards benefit adaptation exploration, we design an environment similar to Point-Robot-Sparse and has a sub-optimal goal that keeps unchanged across all tasks, as shown in Figure 10. While the explorer without intrinsic rewards keeps going to the sub-optimal goal to maximize average return and fails to collect useful information, the intrinsically motivated explorer effectively explores possible goals and manages to find the real goal, thus attaining higher final performance.

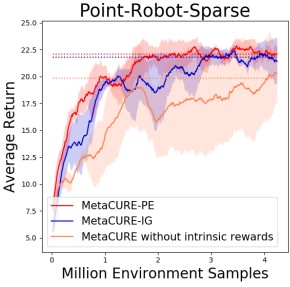

Figure 9: MetaCURE without intrinsic rewards cannot explore effectively during meta-training and performs poorly.

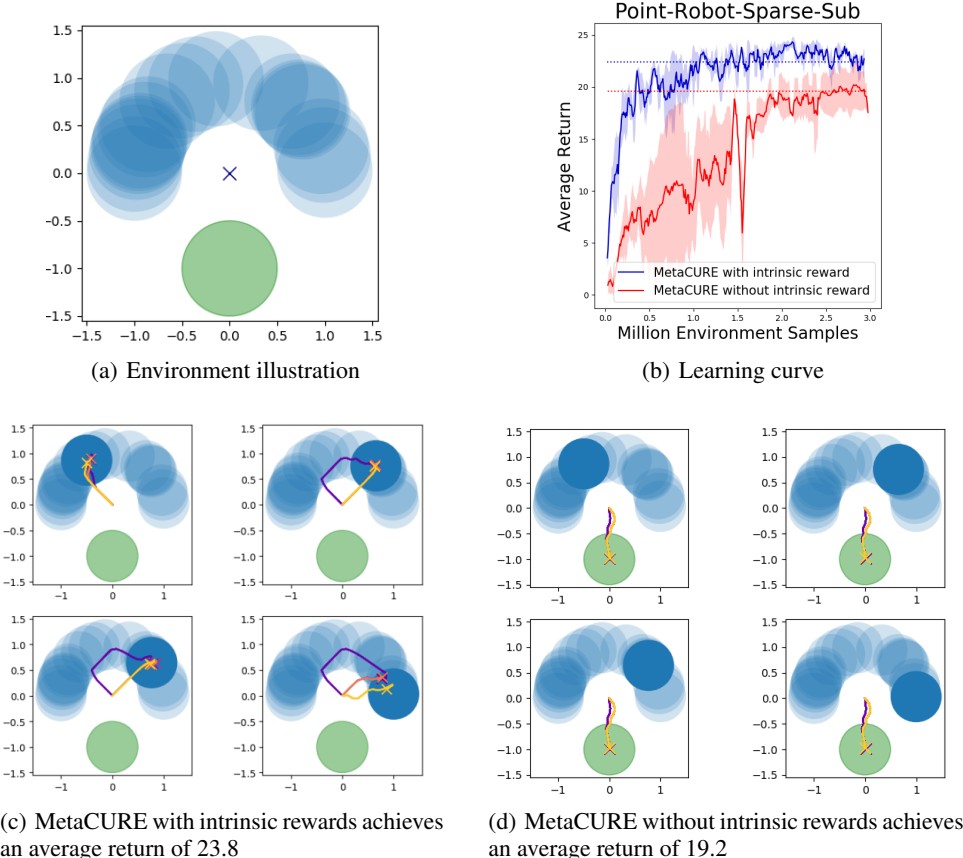

(a) Environment illustration

(b) Learning curve

(c) MetaCURE with intrinsic rewards achieves an average return of 23.8

(d) MetaCURE without intrinsic rewards achieves an average return of 19.2

Figure 10: (a) Point-Robot-Sparse with a sub-optimal goal. The sub-optimal goal is illustrated in green, which requires no adaptation exploration but offers lower returns. The cross indicates the agent's initial position. Blue circles indicate optimal goals. (b) Learning curves of MetaCURE w/o intrinsic rewards on the illustrated task set. (c) Adaptation visualization of MetaCURE with intrinsic rewards (curiosity intrinsic). MetaCURE finds the optimal goal, and leads to higher final performance. (d) Adaptation visualization of MetaCURE without intrinsic rewards. Explorer without intrinsic rewards maximizes average extrinsic return, leading to inefficient exploration.

