# OpenReview forum: "Intrinsically Guided Exploration in Meta Reinforcement Learning"
_ICLR.cc/2021/Conference — Reject_

### Official Review · AnonReviewer4 · 2020-10-21
**Unclear contribution**

**Rating:** 4
**Confidence:** 5

**Review:**

This paper introduces the use of intrinsically motivated exploration in meta-training and adaptation phase in meta Reinforcement Learning. The proposed framework consists of using task-inference, already used in other works like PEARL and VariBAD, and two deep neural networks, trained with an arbitrary off-policy algorithm, respectively for exploitation and exploration phases.

I think the paper is well-written and discusses the related literature sufficiently well. The empirical results are significantly better than the considered baselines.
Despite these good qualities, I think the major weakness of this paper is the lack of contribution. It looks to me that the major contribution discussed by the authors is the use of intrinsically motivated exploration in meta-training and adaptation, while the use of task-inference has been already proposed in other works and the use of different policy networks for exploitation and exploration is trivial. Considering the intrinsically motivated exploration, I miss the importance of the contribution, since it is already well-known that this technique can be beneficial, especially in sparse MDPs. The intrinsic reward makes sense, but it is not novel, as similar intrinsic rewards have been proposed in VIME and Pathak et al. (2017).
Experimental results are good, but not well described. For example, I don't find a clear explanation of the tasks used in meta-training. The random seeds are only three, which makes the results less reliable.
Considering this, I don't consider this paper significant enough to be accepted.

Minor issues:
* Pag. 1, typo: "tasks" -> tasks
* Pag. 1, "Based this objective" -> Based on this objective
* the R function in (1) is not formally defined.

Pros
------
* Well-written;
* Good discussion of related literature;
* Good experimental results.

Cons
-------
* Lack of contribution;
* Few, but important, missing details about the experimental setting, and too few seeds.

Post-rebuttal feedback
-------------------------------
I thank the author for their reply and I encourage them to make the suggested improvements to the paper.

---

> ### Author Response · Authors · 2020-11-23
> **Author Response to Review 4**
>
> Thank you for your comments. We will propose a new version of MetaCURE to overcome its current limitations, which will have a novel objective associated with new intrinsic rewards for meta-RL.

---

### Official Review · AnonReviewer1 · 2020-10-27

**Rating:** 4
**Confidence:** 3

**Review:**

This paper proposes a novel meta-learning method, aimed at solving the following problem: at test time, the agent has N episodes to gather information [exploration phase], and we care about its return in the N+1th episode [exploitation phase]. To this end, the authors propose to learn a separate exploration and exploitation policy. The core of the algorithm is to use an exploration bonus for the exploration policy that rewards finding novel trajectories. This should help it to collect valuable information during meta training and meta testing. The exploration policy is only used in the N+1th episode, and  is conditioned on a context vector which is computed from the exploration trajectories.

Pro:
- Very relevant subject.
- The authors bring up an interesting point, that metaRL methods should consider both exploration during meta training and meta testing.
- I think it makes sense to separate exploration and exploitation policy explicitly.
- The authors propose to sparsify common metaRL benchmarks. Evaluating your algorithm on environments that are proposed in the same paper can be tricky (and often means that the empirical results are evaluated more critically), but I think the suggestion of looking at sparse metaRL tasks is useful for the field and should be considered by other researchers as well.
- The authors provide illustrations of the behaviours of the learned policy at meta-test time, which I think is really nice!

Concerns:
- I fee like there's a mismatch between the reward that the explorer gets, and the problem setting that the authors consider. The way I understand it, the two terms in the explorer's reward (Eq (2)) can be helpful during meta-training but I don't see how that translates to meta-test time. Let me try to explain:
 - [First term] The objective of the agent is to maximise the return in the *last* episode (the exploitation episode). Then why is the explorer not maximising the return of the exploitation phase, but instead gets a reward that corresponds to the exploration episodes? Doesn't that contradict the premise that we're not interested at all in the reward that is collected during exploration? Also, what is n (you write "training iterations" - is that just the number of environment steps per task? number of policy updates over the entirety of meta-training? Does this mean this is annealed to zero over training? Maybe then it shouldn't be called $\gamma$ so that it's not confused with the per-episode discount usually used in RL.).
 - [Second term] The reward bonus can be seen as a type of novelty bonus for finding new trajectories. Can the authors clarify why this would help (a) during meta-training and (b) during meta-testing? I can see how it helps during training, but I don't understand why it also helps at test time. At the end of Sec 3.2, you write "the intrinsic reward is estimated on multiple tasks and does not diminish during meta-training". Why not? In Eq 5, does $c_{-1:i-1}$ come from multiple tasks or just one? Why can't this diminish over time, shouldn't our model $p(r,s|x,a,z)$ (dropping the subscripts for brevity) get better over time? Or do you just mean that the weight $\lambda$ is not annealed? And doesn't it have all the information to, in principle, predict r and s exactly (except for aleatoric noise) so then this term reduces to the noise in the environment? Or should the terms in the subscript not match the ones within the expectation? It's also a bit unclear what the expectation is over, the subscript just shows some tuples. What's the probability distribution? Where do the terms in the subscript come from? Are $q$ and $p^\mathcal{T}$ mixed in here? It would help my understanding if these things could be made more explicit.
 - [Missing term?] I understand the above two terms might help during training, but they are only auxiliary tasks in a way. And if the first term is annealed to zero (if that's indeed what's happening?), and the second term reduces to aleatoric noise (is that's indeed what's happening?), what is the exploration policy even doing at the end of training? Should there be a third term, which is the return in the *exploitation* phase, which is the only thing that the exploration policy should try to maximise at meta-test time?
- I think the mathematical formulation has to be fleshed out. The paper is sometimes hard to follow because the notation is unclear, and not all terms are explained. I have a few open questions because of this:
 - What are the expectations in (3), (4), (5) over?
 - Not all terms in Figure 1 are defined or mentioned in the text. Are $L_\pi$ and $L_{Q_\pi}$ loss terms from SAC, so they correspond to equation (1)? And $L_{\pi_e}$ corresponds to equation (2)? And what is $L_{pred}$? All these should be explicitly defined in the text with an equation.
 - Sec 2, directly after Eq (1): $c^\mathcal{T}_{\pi_e}$ is not defined.
- In Figure 2, PEARL (Walker-Vel-Sparse) and VariBAD (Point-Robot-Sparse) seem to first learn and then diverge. Do the authors have an explanation as to why this is? It seems to me that maybe the hyperparameters just weren't set right, and makes me a bit sceptical of the results. When PEARL/VariBAD *do* learn, their (end-) performance is much closer to that of MetaCURE.
- The authors propose two versions of MetaCURE (IG and PE). On the continuous control tasks PE is better, but then on the noisy point robot PE completely fails and IG succeeds. I think the authors are not doing themselves a favour by setting up the presentation of their methods like this. It would be better to propose *one* method, and then fully focus on this. The way it is now, there are two methods where neither of them is clearly better. I think that's confusing to the reader, and less convincing. Shouldn't there be a way to develop an exploration bonus that can do well across all environments?
- The "noisy" environment is a POMDP (the agent gets noisy state observations). So doesn't that mean that there is simply a mismatch between the problem setting and the way the policy is set up? I'm not sure that this is a good experiment to add, because the solution (using the PE explorer) does not directly address the problem (partial observability). Maybe a noisy reward function would fit better here?

Other questions:
- Can you explain why the same exploration mechanism useful during meta-training and meta-testing? Aren't we mixing across-task and within-task information gain, and if so, why is that okay?
- What is the difference between row 1 and row 2 in Figure 2? This is not explained in the text. Is that separation just because everything in one figure would've been too much? And how exactly were the baselines evaluated, always after the same number of episodes?
- Why is the loss of the explorer not backpropagated through the context encoder? How does that fit together with "our intrinsic reward supports end-to-end learning and collect online data for effective task inference"?
- Why does the exploiter get a sample $z$ and not the entire distribution $q$? Is that because we assume that all information is there and $q$ is concentrated anyway?
- You mention that the intrinsic reward of PE is of lower variance. Could you explain why? Is that a hypothesis (if so, say so explicit) or do you have any empirical/theoretical evidence for this (again, if so, make it explicit).

Suggestions:
- You write that "efficient exploration of meta-RL needs to consider both training and adaptation phases simultaneously". This is a good point and I agree with it, and I think the paper can be improved by fleshing this argument out more. Help the reader understand why this is the case, and what the (different) challenges in exploration are during training and testing. One open question I still have for example (see above) is why the *same* exploration mechanism is useful during meta-training and meta-testing. Why not separate them? This should be explained in the paper. In the related work section, you write that prior methods "utilise policy injected with random noise for meta-training exploration, which is ineffective and empirically fail with sparse rewards". This point can be made about almost all prior methods, and should come in the introduction. Then explain intuitively *why* this fails, maybe which an example. Another question I have is, how does the Ishii et al. 2002 paper back up this point that this distinction exists?
- The experiments section relies too much on the appendix, and it's quite unnatural to read it like this. Either I have to jump back and forth between the appendix and the main text, or I have to take your word for what is shown in the results in the appendix. You introduce five continuous control tasks at the start, but then actually only show 3 and the other 2 are in the appendix. I think this section can be improved by focusing on the main important points in the main text, explain this well and mention that more results can be found in the appendix.
- Mention in the main text that the sparse setting for Meta-World means you use the success signal only. This is a very difficult setting and is worth highlighting instead of burying it in the appendix.
- I think a discussion about what an good/optimal exploration strategy would look like can help the reader understand the problem setting, and what you are trying to achieve. Right now, in Sec 4.3 you write for example that PEARL "explored less effectively" because it keeps the belief about the current task throughout an entire episode. That's not a fundamental problem with Thompson-sampling like methods though, but is quite problem-specific. If the number of adaptation episodes that are available corresponds to the number of possible tasks, then Thompson sampling always does optimal exploration. But, if the number of possible tasks exceeds the number of exploration episodes Thompson sampling fails. Adding discussions like these to understand what type exploration *is* necessary and why your method can achieve it, could strengthen the paper.

Typos:
- Intro, Paragraph 3: taskss -> tasks; Based -> Based on
- Background, Paragraph 2: Full stop missing at end
- 3.2, second paragraph + last paragraph: starts with free space
- Figure 2, caption, writes "five" tasks but there are only three (other two in appendix)
- Sec 4 (intro): 3. What's -> What are; 4. Is MetaCURE's -> Are MetaCURE's



------------------------------------------------------------------------------------
UPDATE

Thanks for your detailed response!

A1: I see, that makes more sense to me now. I'm still not 100% convinced that it might not be better to let the policy entirely meta-learn what a good exploration policy is, and only use the inductive bias (terms 1 and 2) for how to do so for meta-training and anneal those terms over time. This would mean that at test time the only thing that the exploration policy should do is maximise the return in the *exploitation* phase - and basically figure out what the best way to do so is (so you'd need the "missing term" I describe above). But that's just a hunch, not sure if that would actually work better.

A2-A8: Thanks a lot for clarifying.

I'll keep my current score given all reviewers agree that the work is unpolished in its current form, and because the authors plan to propose a new version of MetaCURE soon. I think this is very interesting and promising work and look forward to reading an updated version in the future!

---

> ### Author Response · Authors · 2020-11-23
> **Author Response to Review 1 (Part 1)**
>
> Thank you for your comments. A new version of MetaCURE will be proposed soon to improve novelty as well as performance.
>
> Q1: The way I understand it, the two terms in the explorer's reward (Eq (2)) can be helpful during meta-training but I don't see how that translates to meta-test time. \& Can you explain why the same exploration mechanism useful during meta-training and meta-testing?
>
> A1: Our intrinsic reward accounts for exploration in both meta-training and adaptation. The first term of Equation 2 is the extrinsic reward, and is annealed to zero during meta-training. The second term relies on the context $c_{-1:i-1}$, which are experiences collected from one task. Note that the intrinsic reward diminishes only if we have an exact task embedding $z$, which must be inferred from the current context $c_{-1:i-1}$. When the agent tries to adapt to a new task, we do not have enough experiences to obtain an exact $z$, and the intrinsic reward is thus not only aleatoric noise, but also considers task uncertainty given the current context. As for meta-training exploration, a simple insight is that the prediction error serves as a kind of curiosity, and encourages exploration in meta-training.
>
> Although we can directly optimize an exploration policy in order to maximize the exploitation return, this can be extremely difficult with sparse rewards. MetaCURE addresses this problem by developing an intermediate objective for exploration.
>
> Q2: The paper is sometimes hard to follow because the notation is unclear, and not all terms are explained.
>
> A2: The expectation term $ E_{(z,r_i,s_{i+1)}|(c_{-1:i-1},a_i)}$  in Equation 3, 4 and 5 represents an expectation over $p(z|c_{-1:i-1},a_i)$ and $p(r_i,s_{i+1}|c_{-1:i-1},a_i)$. The first term can be estimated by the context encoder $q_\phi(z|c)$ (the action can be omitted [1]), and the second term is achieved by sampling from the replay buffers $B$ and $B_{enc}$.
>
> $L_\pi$, $L_{Q_{\pi}}$, $L_{\pi_e}$ and $L_{Q_{\pi}^e}$ are corresponding SAC loss functions ($L_{Q_{\pi}^e}$ was omitted in Figure 2 for brevity). $L_{pred}$ corresponds to $r_{PE}'$ in Equation 5. To be exact,
>
> $
> L_{Q_{\pi}}=E_{(s,a,r,s')\sim B, c\sim B_{enc},    z\sim q_{\phi}(z|c)}\left[Q_{\pi}(s,a,z)-(r+\gamma\overline{V_{\pi}}(s',\overline{z}))\right]^2
> $
>
> $L_{\pi}=E_{s\sim B,a\sim\pi , c\sim B_{enc}, z\sim q_{\phi}(z|c)}\left[D_{KL}(\pi(a|s,\overline{z})||\frac{exp(Q_\pi(s,a,\overline{z}))}{Z_\pi(s)})\right]$
>
> $L_{Q_{\pi}^e}=E_{(s,a,r,s')\sim B, c\sim B_{enc}}\left[Q_{\pi^e}(s,a,\overline{q_\phi}(z|c))-(r_{explorer}+\gamma\overline{V_{\pi^e}}(s',\overline{q_\phi}(z|c'))\right]^2$
>
> $L_{\pi^e}=E_{s\sim B,a\sim\pi , c\sim B_{enc}}\left[D_{KL}\left(\pi^e(a|s,\overline{q_\phi}(z|c))||\frac{exp(Q_\pi^e(s,a,\overline{z}))}{Z_{\pi^e}(s)}\right)\right]$
>
> , where $c'=c\cup \{(s,a,r,s')\}$ is the updated context. $Z_\pi(s)$ and $Z_{\pi^e}(s)$ are normalization functions that do not affect gradients. $\overline{V_{\pi}}$ and $\overline{V_{\pi^e}}$ are target value functions. $\overline{z}$ and $\overline{q_\phi}$ indicate that gradients do not flow through them.
>
> $c_{\pi_e}^\mathcal{T}$ is defined in the last paragraph of Section 2.
>
> Q3: In Figure 2, PEARL (Walker-Vel-Sparse) and VariBAD (Point-Robot-Sparse) seem to first learn and then diverge. \& What is the difference between row 1 and row 2 in Figure 2?
>
> A3: In Walker-Vel-Sparse, PEARL eventually converges, but cannot perform as well as MetaCURE due to its inefficient adaptation exploration. Point-Robot-Sparse requires powerful meta-training exploration, and VariBAD lacks such a metra-training exploration mechanism, which makes it unstable.
>
> We separated the training curves into two figures to make it clearer to read. The baselines are evaluated after the same number of meta-training samples, as mentioned in Section 4.1.
>
> Q4: The authors propose two versions of MetaCURE (IG and PE)...It would be better to propose one method, and then fully focus on this.
>
> A4: We will propose a new objective for MetaCURE, from which we can derive a novel intrinsic reward that avoids the drawbacks of the current intrinsic rewards proposed in this work.
>
> Q5: The "noisy" environment is a POMDP (the agent gets noisy state observations).
>
> A5: The noisy environment is actually not a POMDP. As stated in Section 4.4, the agent's $(x,y)$ position is not disturbed by noise, and noise is only added to a state variable that is irrelevant to decision making. This setting satisfies the Markov property: given current observation and action, the next observation's probability distribution is fully determined.
>
> [1] Houthooft, Rein , et al. "VIME: Variational Information Maximizing Exploration." (2016).

---

> > ### Author Response · Authors · 2020-11-23
> > **Author Response to Review 1 (Part 2)**
> >
> > Q6: Why is the loss of the explorer not backpropagated through the context encoder?
> >
> > A6: Empirically we find that only backpropagating the exploitation policy's critic loss through the context encoder makes learning efficient and robust. The algorithm can still be trained end-to-end by using the stop-gradient operators.
> >
> > Q7: Why does the exploiter get a sample $z$ and not the entire distribution $q$?
> >
> > A7: The exploiter gets a sample $z$ because we assume that the agent has got sufficient information and the posterior has converged to nearly a point estimate.
> >
> > Q8: You mention that the intrinsic reward of PE is of lower variance. Could you explain why?
> >
> > A8: The PE intrinsic reward obtains lower variance because it is estimated with a single neural network, while the IG intrinsic reward is estimated with two neural networks, which is of higher variance. Learning curves in Section 4.1 also support our claim.

---

### Official Review · AnonReviewer2 · 2020-10-28
**Interesting approach but concerns regarding experiments and novelty**

**Rating:** 4
**Confidence:** 4

**Review:**

Summary: This paper aims to facilitate meta-reinforcement learning by improving sample efficiency during meta-training as well as adaptation. Towards this goal, the paper proposes to use an intrinsic reward (two versions) and to learn a separate exploration policy to collect data for faster task inference.

Strengths -

Approaches to improve sample efficiency in meta-RL would be of interest to the community and are potentially useful to RL practitioners.

The proposed technique does not utilise dense rewards during meta-training. The use of intrinsic rewards in this approach possibly helps to overcome an important weakness of some previous methods (like PEARL or MAESN), which find it difficult to use sparse rewards during meta-training and assume access to dense reward in the meta-training phase.

The approach outperforms other meta-RL techniques within the given experimental setup. Additionally, the visualisations of adaptation behaviour (like in Figures 3 and 4) were quite helpful in understanding the method. However, I will discuss my concerns regarding the experiments in the subsequent sections.

Weaknesses and concerns -

The writing and clarity of the paper can be improved. It could be useful to provide additional details to motivate the problem and choice of experimental setting. In the introduction, providing further comments (or citations) could justify this claim: “However, these exploration mechanisms are still insufficient in either meta-training or adaptation and underperform in complex sparse-reward tasks.” It would be helpful to provide more details about the proposed method beyond Figure 1. Is the encoder a recurrent network? And the meta-predictor is another feedforward neural network?

Regarding the experiments, I would like to better understand the motivation behind the choice of restricting the adaptation phase to 2-4 episodes and then evaluating on the last episode. Is this choice realistic in applications? It could be worth considering that such a choice does not take into account potentially catastrophic behaviour by the exploratory policy and is closer to best arm identification (rather than regret minimization) in the multi-armed bandit setting.

It appears that a crucial reason for the better performance of the proposed approach is that it can explore within an episode. The off-policy competitor, PEARL, which uses posterior sampling, keeps its belief fixed for one episode and takes actions accordingly. While the experiments illustrate this weakness of the specifics of PEARL, in principle, there is no requirement in posterior sampling to update and sample from the posterior only once per episode. Since using an intrinsic reward introduces new hyperparameters, it could also be possible to introduce a hyperparameter for the number of steps (say T) between which PEARL updates and samples from the posterior. Tuning such a hyperparameter will also allow PEARL to explore within an episode and could make it much harder to beat. It could be informative to see such a comparison.

Another main concern is regarding the novelty of the proposed approach since the major off-policy meta-RL component appears similar to PEARL. And as the authors mention in the related work section, similar intrinsic rewards and maintaining a separate exploration policy (Gurumurthy et al., 2019) have also been studied earlier.

Related work -

The related work section is missing the original references. For example, the paper refers to recent work related to the use of prediction error as an intrinsic reward: Oh et al. (2015), Stadie et al. (2015) and Pathak et al. (2017). However, prediction error-based intrinsic rewards were introduced in 1990 [b]. Using information gain as a reward for RL agents to better explore the environment was introduced in 1995 [a]. Meta-RL was introduced in 1994 [c]. Intrinsically guided exploration in Meta-RL was introduced in 1997-2002 [d]. Please correct and describe what's really novel in your approach.

[a] J. Storck et al. "Reinforcement driven information acquisition in non-deterministic environments." Proceedings of ICANN'95, Paris. Vol. 2. 1995.

[b] J. Schmidhuber. "A possibility for implementing curiosity and boredom in model-building neural controllers." Proc. of SAB 1991.

[c]          ---.          "On learning how to learn learning strategies." TR FKI-198-94, TUM, 1994. http://people.idsia.ch/~juergen/FKI-198-94ocr.pdf

[d]          ---.          "What's interesting?" TR IDSIA-35-97, IDSIA, July 1997. Also in: Exploring the Predictable. In Ghosh, S. Tsutsui, eds., Advances in Evolutionary Computing, p. 579-612, Springer, 2002. ftp://ftp.idsia.ch/pub/juergen/explorepredictable.pdf


Other minor comments -

- At first glance, it was slightly unclear what the curves in Figure 2 represent. I would suggest moving something like the statement on Page 4, “meta-testing performance as a function of number of experiences collected during meta-training” to the caption of Figure 2.


Some typos -

- Page 1, paragraph 3 in the Introduction:

taskss -> tasks
Based this objective -> Based on

- Page 2, section 3.1, 1st paragraph

Explorer policy to learn an behavior -> a behavior

- Page 3, last paragraph

KL distance -> divergence

- Page 7, Figure 5 caption

nosier -> noisier

---

> ### Author Response · Authors · 2020-11-23
> **Author Response to Review 2**
>
> Thank you for your comments. A new version of MetaCURE will be proposed soon to improve novelty as well as performance.
>
> Q1: It would be helpful to provide more details about the proposed method beyond Figure 1.
>
> A1: The encoder can be any network that effectively deals with sequence inputs, such as RNNs. In the implementation, we utilize SNAIL [1] as our encoder. The meta-predictor is a feedforward neural network.
>
> Q2: Regarding the experiments, I would like to better understand the motivation behind the choice of restricting the adaptation phase to 2-4 episodes and then evaluating on the last episode. Is this choice realistic in applications?
>
> A2: The length of the adaptation phase can vary with the specific task set. In application, our method can be applied to scenarios that do not consider exploration costs, such as adapting to new computer games or chess games. Best arm identification shares some similarity with the meta-RL objective, but is restricted to bandit problems.
>
> Q3: Since using an intrinsic reward introduces new hyperparameters, it could also be possible to introduce a hyperparameter for the number of steps (say T) between which PEARL updates and samples from the posterior. Tuning such a hyperparameter will also allow PEARL to explore within an episode and could make it much harder to beat. It could be informative to see such a comparison.
>
> A3:
> We test updating PEARL every 10 steps on Point-Robot-Sparse, and its performance rises to $16.2\pm 2.3$, better than the original PEARL ($11.2\pm 1.6$) but still underperforms MetaCURE ($24.2\pm 0.4$). Also, the hyperparameter $T$ needs careful finetuning for every task set, while in MetaCURE the intrinsic reward weight does not need extra tuning.
>
> Q4: Another main concern is regarding the novelty of the proposed approach.
>
> A4: We will propose a new version of MetaCURE, in which we propose a novel intrinsic reward that avoids the drawbacks of the two old variants.
>
> [1] Mishra, Nikhil , et al. "A Simple Neural Attentive Meta-Learner." (2017).

---

### Official Review · AnonReviewer3 · 2020-10-28
**Promising direction but unpolished work**

**Rating:** 4
**Confidence:** 3

**Review:**

*SUMMARY*

The paper presents a method for efficient task identification to improve adaptation in a meta RL setting. The approach is based on learning an exploration policy to quickly discriminate the task at hand, so that to leverage a task-specific policy for exploitation. To do so, it employs an intrinsic reward proportional to the information gain (or prediction error) over both the transition and reward models. Finally, the algorithm is evaluated over a set of continuous control domains with sparse rewards.

*EVALUATION*

The idea of focusing the exploration on identifying the current task, instead of collecting good samples for policy improvements, is quite interesting, albeit not completely novel. Unfortunately, the lacking presentation does not help assessing the real quality of the contributions and the robustness of the experimental analysis, and it make me leaning towards a negative evaluation. However, I believe that a partial overhaul of the paper structure, along with a sharper experimental analysis, would make for a nice contribution to the meta RL community.

*DETAILED COMMENTS*

C1) As noted by the authors, the idea of maximizing information gain over the transition model in a single-environment setting has been presented in VIME (Houthooft et al. 2016), while Zhou et al. (2018) employs information gain to fast identification of the transition model in multi-environment setting. Thus, the main contribution of this work seems to account also for the reward model in the information gain computation. However, it is not clear to me how this can lead to a significant improvement, especially in sparse-reward settings, where rewards are zero almost everywhere.

C2) An information theoretic approach to leverage the reward structure in a multi-goal scenario has been previously proposed in (Goyal et al. InfoBot: Transfer and exploration via the information bottleneck. ICLR 2019). Though the setting is not equivalent, I am  wondering how their concept of *decision state* relates to the idea of fast identification of the reward model.

C3) The paper propose two alternatives reward function, one that essentially normalizes the prediction error with the predictability of the transition (IG), and the other that solely consider the prediction error (PE). While the experimental analysis shows a slight edge for the latter, I would say that this might be negligible with respect to the ability of the former to deal with noisy environments (as explained in Section 4.4).

*QUESTIONS*

- Could the authors address the comments above in their response?
- The last line of Section 3 says that the proposed intrinsic reward does not vanish over time: Can the authors explain why?
- The experimental setup describes a very short adaptation phase (hundreds of samples) but the subsequent plots are functions of millions of samples. Could the authors clarify the experimental setup, especially the number of samples employed in the meta-training and meta-testing phases?


*ADDITIONAL FEEDBACK*

- I would suggest to revise the structure of the paper by focusing on the core ideas first, and to the implementation details later.
- I did not find the notation crystal clear, especially regarding the role of the context c, and the indiscriminate use of the words trajectory, episode, experience. To me it would be better to refer as sample every (s,a,r,s) tuple and trajectory every sequence of samples from the initial state to the end of an episode.
- In my opinion the experimental section could be sharper. I would suggest to focus on a smaller set of domains (such as one with changing dynamics, one with changing rewards and one with both for the Mujoco set, in addition to the two Meta-World) and a smaller set of comparisons, and to seek for a clearer illustrations of the benefits of MetaCURE with respect to other methods.
- I believe the ablation study to be central in the evaluation of MetaCURE. Especially, I would have preferred to see a convincing assessment of the benefit of computing the information gain over both the reward and transition models, with respect to accounting just for the former or the latter.
- I would not claim the performance of Meta-World Push (Table 1) to be significantly higher than PEARL, since the confidence intervals are overlapping. Averaging over more trials might help.
- I think the main text relies too much on the Appendix: It is fine to have proofs and derivations out of the paper, but I would not directly comment in the experimental section figures that only appear in the Appendix.

#############

AFTER RESPONSE

I would like to thank the authors for the detailed response. I encourage them to keep working on MetaCURE: With some improvements I believe it will provide a valuable contribution to the meta-RL field.

---

> ### Author Response · Authors · 2020-11-23
> **Author Response to Review 3**
>
> Thank you for your comments. A new version of MetaCURE will be proposed soon to improve novelty as well as performance.
>
> Q1: The main contribution of this work seems to account also for the reward model in the information gain computation. However, it is not clear to me how this can lead to a significant improvement, especially in sparse-reward settings, where rewards are zero almost everywhere.
>
> A1: By considering information gain in reward model, MetaCURE encourages exploration on regions that potentially obtain non-zero rewards, which can be much larger than rewarding regions in a single task. As shown in Figure 3 in original paper, in sparse-reward settings, although the rewarding region for a single task is small (dark blue circle),  the potentially rewarding region can be large (light blue circles).
>
> Q2: I am wondering how their concept of `decision state' relates to the idea of fast identification of the reward model.
>
> A2: InfoBot captures invariances shared across goal-conditioned policies, and 'decision states' refer to states where policies differ greatly and need extra exploration when transferring to new goals. In contrast, MetaCURE finds states where the rewards and transitions differ across tasks, and encourages visitation to these informative states to improve task inference.
>
> Q3: While the experimental analysis shows a slight edge for the latter (MetaCURE-IG), I would say that this might be negligible with respect to the ability of the former to deal with noisy environments.
>
> A3: In the next version of MetaCURE, we will propose a new objective function, from which novel intrinsic rewards will be derived. This new intrinsic reward will be both stable and noise-robust, avoiding the shortcomings of both MetaCURE-IG and MetaCURE-PE.
>
> Q4: The last line of Section 3 says that the proposed intrinsic reward does not vanish over time: Can the authors explain why?
>
> A4: As shown in Equation 4 and 5, the intrinsic reward relies on $z$, and diminishes if $z$ is accurate and we have an accurate predictor. However, when adapting to a new task, we do not have enough experiences to acquire a good estimate of $z$, and the intrinsic reward is thus not zero.
>
> Q5: Could the authors clarify the experimental setup, especially the number of samples employed in the meta-training and meta-testing phases?
>
> A5: As stated in the `Results and analyses' part in Section 4.1, the horizontal axis represents number of meta-training examples, and the vertical axis is the final performance after few-shot adaptation. To be exact, we first meta-train the agent (with millions of samples), and then test its adaptation performance with few samples.

---

### Author Response · Authors · 2020-11-23
**Thank you for your helpful comments. We will propose a new version of MetaCURE.**

Dear reviewers:

Thank you for your inspiring comments.

We will propose a new version of MetaCURE, in which we derive a novel objective for MetaCURE, and overcome the limitations of variants proposed in the current version. We also revise the paper's structure and improve writing, to make it clear and easy to follow. We believe that these improvements will help MetaCURE become a more mature work.

---

### Decision · Program_Chairs · 2021-01-07
**Final Decision**

**Decision:**

Reject

**Comment:**

The paper proposes a novel off-policy meta-RL algorithm able to achieve efficient exploration in meta-training able to perform a fast task identification.
Although the reviewers agree that this paper has merits (relevant topic, interesting idea, nice experimental analysis), they have raised several concerns about the clarity, the novelty, and the soundness of the proposed approach, which make the paper not ready for publication. However, the authors are encouraged in improving their approach since the direction is considered promising by the reviewers.